# Checkpoint Defects Elicit a WRNIP1-Mediated Response to Counteract R-Loop-Associated Genomic Instability

**DOI:** 10.3390/cancers12020389

**Published:** 2020-02-07

**Authors:** Veronica Marabitti, Giorgia Lillo, Eva Malacaria, Valentina Palermo, Pietro Pichierri, Annapaola Franchitto

**Affiliations:** Department of Environment and Health, Section of Mechanisms Biomarkers and Models, Istituto Superiore di Sanita’, Viale Regina Elena 299, 00161 Rome, Italy; veronicamarabitti1@gmail.com (V.M.); giorgia.lillo02@gmail.com (G.L.); evamalacaria@gmail.com (E.M.); vale.palermo89@libero.it (V.P.); Pietro.pichierri@iss.it (P.P.)

**Keywords:** replication stress, DNA repair, genomic instability

## Abstract

Conflicts between replication and transcription are a common source of genomic instability, a characteristic of almost all human cancers. Aberrant R-loops can cause a block to replication fork progression. A growing number of factors are involved in the resolution of these harmful structures and many perhaps are still unknown. Here, we reveal that the Werner interacting protein 1 (WRNIP1)-mediated response is implicated in counteracting aberrant R-loop accumulation. Using human cellular models with compromised Ataxia-Telangiectasia and Rad3-Related (ATR)-dependent checkpoint activation, we show that WRNIP1 is stabilized in chromatin and is needed for maintaining genome integrity by mediating the Ataxia Telangiectasia Mutated (ATM)-dependent phosphorylation of Checkpoint kinase 1 (CHK1). Furthermore, we demonstrated that loss of Werner Syndrome protein (WRN) or ATR signaling leads to formation of R-loop-dependent parental ssDNA upon mild replication stress, which is covered by Radiorestistance protein 51 (RAD51). We prove that Werner helicase-interacting protein 1 (WRNIP1) chromatin retention is also required to stabilize the association of RAD51 with ssDNA in proximity of R-loops. Therefore, in these pathological contexts, ATM inhibition or WRNIP1 abrogation is accompanied by increased levels of genomic instability. Overall, our findings suggest a novel function for WRNIP1 in preventing R-loop-driven genome instability, providing new clues to understand the way replication–transcription conflicts are handled.

## 1. Introduction

DNA damage or unusual DNA structures may pose a serious impediment to DNA replication, thus threatening genome integrity. One of the major obstacles to the replication fork progression is transcription [1,2]. The main transcription-associated structures that can be detrimental to fork movement are R-loops [3,4]. They are transient and reversible structures forming along the genome, consisting of a DNA–RNA hybrid and a displaced single-stranded DNA. Despite their beneficial function in a series of physiological processes, such as transcription termination, regulation of gene expression, and DNA repair [5], they can cause a head-on clash between the replisome and the RNA polymerase, leading to R-loop-driven replication stress if their turnover is deregulated [6,7]. Therefore, R-loop accumulation is a leading source of replication stress and genome instability. Because critical levels of R-loops may contribute to the heightened cancer predisposition of humans, cells have evolved multiple factors to prevent/remove these harmful structures. Apart from direct regulators of R-loop levels, such as ribonucleases and RNA–DNA helicases, a lot of evidence has highlighted the importance of several replication fork protection factors and DNA damage response (DDR) proteins in counteracting pathological R-loop persistence. Among them are breast cancer gene 1 product (BRCA1) and 2 [8,9,10], the components of the Fanconi Anemia pathway [11,12,13], DNA helicases RECQ5 and BLM [14,15], and the apical activator of the DDR, the ATM kinase [16]. Interestingly, defects in the ATR-CHK1 signaling promote accumulation of aberrant R-loops [17]. Recently, a crucial function in the response to R-loop-associated genome instability in human cells has been reported for the Werner syndrome protein (WRN) [18]. The Werner syndrome (WS) is a severe human disease, caused by genetic mutations that lead to loss of WRN protein, and as a result, affected individuals are predisposed to the early-onset of several cancer types [19,20]. WRN is a protein belonging to the RecQ family of DNA helicases essential in genome stability maintenance, with a role in the repair and recovery of stalled replication forks [21,22]. WRN-deficient cells show impaired ATR-dependent checkpoint activation after short times of treatment with aphidicolin-induced mild replication stress (MRS) [23]. Among the plethora of WRN-interacting proteins participating in the maintenance of genome stability, there is WRNIP1 [24,25]. WRNIP1 is a member of the AAA+ class of ATPase family that is evolutionary conserved [24,26]. Although the yeast homolog of WRNIP1, maintenance of genome stability 1 (Mgs1), is required to prevent genome instability caused by replication arrest [27], little is known about the function of human WRNIP1. Previous studies established that WRNIP1 binds to forked DNA resembling stalled forks [28]. More recently, we demonstrated that WRNIP1 is recruited to hydroxyurea-induced stalled replication forks interacting with RAD51 [29]. Indeed, WRNIP1 is directly involved in preventing uncontrolled MRE11-mediated degradation of stalled forks by promoting RAD51 stabilization on single-stranded DNA (ssDNA) [29]. Furthermore, WRNIP1 has been implicated in the activation of the ATM-dependent checkpoint in response to MRS [30].

In this study, we explore a role of WRNIP1 in limiting R-loop-associated genomic instability upon MRS in cells with compromised ATR-mediated checkpoint response.

## 2. Results

### 2.1. Combined Loss of WRNIP1 and WRN Results in Increased Sensitivity of Cells to MRS

WRNIP1 was originally identified as a WRN-interacting protein [24], but there is no evidence that they cooperate in response to MRS. Hence, we first investigated if WRN and WRNIP1 interact in vivo by testing their coimmunoprecipitation. To this aim, HEK293T cells were transfected with an empty vector or the FLAG-tagged wild-type WRNIP1 and treated or not with a low dose of aphidicolin (Aph). Under untreated conditions, WRNIP1 and WRN coimmunoprecipitated, as expected, and Aph slightly increased this interaction (Figure 1A). This result supports a possible cooperation of WRNIP1 and WRN in response to MRS.

WRN-deficient cells are hypersensitive to Aph [31]. Hence, to gain insight into the role of WRNIP1 and WRN in maintaining genome stability upon MRS, we examined the level of chromosomal damage in cells lacking both proteins. MRC5SV cells stably expressing WRNIP1-targeting shRNA (shWRNIP1) [29] and their parental, wild-type, counterpart (MRC5SV) were transiently transfected with WRN-targeting siRNA. After transfection, cells were exposed to Aph and aberrations scored in metaphase chromosomes. As previously reported [31], depletion of WRN determined an increase in the average number of gaps and breaks both in unperturbed and Aph-treated samples, with respect to wild-type cells (Figure 1B and Appendix A). On the contrary, shWRNIP1 cells presented a level of chromosomal damage that is slightly higher than in wild-type cells, but lower with respect to WRN-depleted cells (Figure 1B). Notably, concomitant depletion of WRNIP1 and WRN was associated with a synergistic enhancement of the chromosomal aberration frequency (Figure 1B).

In parallel experiments, we also evaluated the presence of DNA damage by alkaline Comet assay. Loss of WRNIP1 or WRN led to higher spontaneous levels of DNA damage compared to wild-type cells, and depletion of WRN in shWRNIP1 cells significantly increased DNA damage accumulation with respect to each single deficiency (Figure 1C). Similarly, concomitant depletion of WRN and WRNIP1 increased DNA damage above the levels observed in single-depleted cells after Aph treatment (Figure 1C).

Overall, these findings indicate that WRNIP1-deficiency sensitizes cells to Aph and that combined loss of WRNIP1 and WRN exacerbates this sensitivity. Furthermore, they suggest that these two proteins do not function cooperatively in response to MRS, even if Aph stimulates their association in a complex.

### 2.2. WRNIP1 is Tightly Associated with Chromatin in WS Cells

Our results argue that WRNIP1 and WRN cooperate to maintain genome integrity upon MRS, although they do not act in the same pathway. Hence, we verified whether WRNIP1 might be required in the absence of WRN. Because protein recruitment/retention onto chromatin is a critical process for DNA metabolism, we monitored the chromatin association of WRNIP1 in WRN-deficient cells (WS) and in isogenic WRN wild-type corrected (WSWRN) counterpart. Cells were treated or not with Aph and subjected to a chromatin fractionation assay at increasing concentrations of NaCl combined with detergent pre-extraction (Figure 2A). WRNIP1 total levels were comparable in WS and WS-corrected cell lines under unperturbed and Aph-treatment conditions (Figure 2B). Although low salt extraction did not greatly influence WRNIP1 binding to chromatin, under high salt concentration, the chromatin-bound fraction of the protein was higher in WS than in WS-corrected cells (Figure 2B). Similar results were obtained in cells transiently depleted of WRN (WRN-kd), indicating that increased stability of WRNIP1 in chromatin is not cell type-dependent but related to the absence of WRN (Appendix A). Our findings suggest that loss of WRN increases the affinity of WRNIP1 for chromatin.

### 2.3. WRNIP1 Mediates ATM Signaling Activation Leading to CHK1 Phosphorylation in Response to MRS in WS Cells

WRNIP1 promotes ATM signaling activation upon MRS [30], and we recently reported a hyperactivation of ATM in WRN-deficient cells after mild replication stress [18]. To elucidate the significance of the higher chromatin-affinity of WRNIP1 in WS cells, we evaluated whether it correlates to ATM activation. Hence, we analyzed by immunofluorescence the presence of the phosphorylation of ATM at Ser1981 (pATM), an accepted marker for ATM activation [32], in wild-type (WSWRN) or WS cells depleted of WRNIP1 treated or not with Aph. As expected, pATM levels were increased in the absence of WRN (Figure 2C). Notably, downregulation of WRNIP1 significantly reduced the strong ATM activation in WS cells (Appendix A and Figure 2C), confirming that WRNIP1 is required in establishing the idiosyncratic ATM signaling associated with loss of WRN [18].

Next, we tested whether the WRNIP1-mediated ATM pathway could participate in late CHK1 activation observed in WS cells [18]. To this aim, WS cells were depleted for WRNIP1 and treated or not with Aph. Loss of WRNIP1 compromised CHK1 activation after treatment in WS cells, and the reduction of phospho-CHK1 levels was similar to that caused by ATM inhibition (Figure 2D). Consistently with a role of WRNIP1 in activating an ATM signaling, combined loss of WRNIP1 and ATM did not have any additive effect on CHK1 phosphorylation (Figure 2D). Absence of WRN differently affects CHK1 activation upon MRS: it reduces the ATR-dependent CHK1 activation early after Aph treatment but stimulates that dependent on ATM at late time-points [18]. The two phenotypes are interlinked and expression of a phospho-mimic form of CHK1, CHK1^317D/345D^ [33], prevents the late phenotype [18]. Hence, we verified whether expression of the phospho-mimic mutant of CHK1 could hinder stable association of WRNIP1 with chromatin in WS cells. Notably, a normal binding of WRNIP1 to chromatin was restored by expression of the phospho-mimic CHK1 mutant in WS cells (Figure 2E). This suggests that WRNIP1 is required for an ATM-mediated activation of CHK1 and that its stable recruitment in chromatin is triggered by the impaired early CHK1 phosphorylation.

To reinforce this hypothesis, we investigated the binding of WRNIP1 to chromatin in WRN^K577M^ cells, which efficiently phosphorylate CHK1 early after Aph [23]. Fractionation analysis showed that the level of chromatin-bound WRNIP1 in WRN^K577M^ cells was lower than that in WS cells (Appendix A). However, following CHK1 inhibition by UCN-01 [34], the amount of chromatin-associated WRNIP1 was greatly enhanced in WRN^K577M^ cells as well as in wild-type cells (Appendix A).

Therefore, our findings suggest that, in WRN-deficient cells, WRNIP1 is strongly associated with chromatin and related to ATM-dependent CHK1 phosphorylation in response to MRS.

### 2.4. Depletion of Essential Factors for Activation of ATR-CHK1 Pathway Promotes WRNIP1 Retention in Chromatin

Increased stability of WRNIP1 in chromatin is linked to defective CHK1 activation observed in the absence of WRN early after MRS. Several factors facilitate the early ATR-mediated CHK1 phosphorylation, including the ATR kinase-activating protein DNA Topoisomerase 2-binding protein 1 (TopBP1) [35] and the CHK1-interacting factor Claspin [36,37]. Hence, we asked whether increased binding of WRNIP1 to chromatin is a general response to compromised phosphorylation of CHK1. To this aim, we used siRNAs to deplete endogenous TopBP1 or Claspin expression in wild-type cells (WSWRN) and examined WRNIP1 retention in chromatin at high salt concentration upon MRS. The total amount of WRNIP1 was comparable in mock-depleted, TopBP1-, or Claspin-depleted cells under unperturbed conditions or after treatment (Figure 3A). Interestingly, however, TopBP1 or Claspin depletion greatly enhanced WRNIP1 binding to chromatin under high salt concentration, irrespective of the Aph treatment (Appendix A and Figure 3A).

Loss of WRN leads to a significant induction of phospho-ATM upon MRS in wild-type cells [18]. We therefore evaluated the phosphorylation of ATM in wild-type cells depleted of TopBP1 or Claspin. Staining against pATM showed that, after depletion of TopBP1 or Claspin, ATM was hyper-phosphorylated and that Aph significantly increased its activation (Figure 3B). Interestingly, no significant differences were noted using hydroxyurea (HU) as replication-perturbing treatment (Figure 3B), suggesting that this response is specific of an MRS that does not arrest completely replication fork progression.

Next, we explored whether the ATM pathway is involved in activating CHK1 in TopBP1-depleted cells as observed in WS cells. Our results showed that, although Aph activated CHK1 in both cell lines, CHK1 phosphorylation was only modestly reduced by ATM inhibition in wild-type cells but appeared considerably hampered in TopBP1-depleted cells (Figure 3C). As expected, CHK1 was not phosphorylated after short-term exposure to Aph in the absence of TopBP1 (Appendix A).

Collectively, these results indicate that impairment of the early ATR-dependent CHK1 activation after MRS calls for increased stability of WRNIP1 in chromatin. Furthermore, they suggest that a WRNIP1-mediated ATM signaling is hyperactivated whenever CHK1 phosphorylation is defective.

### 2.5. Combined Abrogation of ATM Activity and WRNIP1 Enhances DNA Damage after MRS in Cells with Defective ATR-CHK1 Signalling

Our data indicate that MRS triggers a WRNIP1-mediated ATM signaling in cells with impaired CHK1 phosphorylation. However, apart from its role in promoting an ATM pathway, WRNIP1 acts as a replication fork protection factor [29]. Moreover, ATM inhibition exacerbates the already elevated genomic instability in WRN-deficient cells [18]. Hence, we asked whether abrogating the ATM pathway and WRNIP1 function could exacerbate genome instability further in cells with defective ATR-CHK1 signaling in response to MRS. As a model of impaired early activation of CHK1, we used WS cells or cells depleted of TopBP1 or Claspin. We first performed the alkaline Comet assay in WS cells treated with ATM inhibitor and/or siRNA against WRNIP1. As expected, in the absence of WRN, depletion of ATM or WRNIP1 markedly increased the extent of DNA damage after Aph and concomitant depletion of ATM/WRNIP1 further strengthened it (Figure 4A and Appendix A).

Similarly, the downregulation of TopBP1 or Claspin in WSWRN cells resulted in DNA damage potentiation upon MRS (Figure 4B). Notably, more than an additive effect was detected after a combination of ATM inhibition and TopBP1 or Claspin depletion in Aph-treated wild-type cells (Figure 4B).

Of note, treatment with Aph induced a significantly higher level of DNA breakage in cells with concomitant loss of ATM/WRNIP1 than in cells depleted of WRNIP1 alone (Figure 4C and Appendix A). However, upon MRS, the amount of DNA damage was even further increased in wild-type cells with triple TopBP1/ATM/WRNIP1 abrogation in comparison with double knockdown (Figure 4C).

Therefore, combined loss of ATM activity and WRNIP1 potentiates DNA damage after MRS in cells with defective ATR-CHK1 signaling. This suggests that WRNIP1 may play an additional role beyond its function in activating an ATM pathway.

### 2.6. Retention of WRNIP1 in Chromatin Correlates with the Presence of RAD51 in WS Cells

We have previously demonstrated that WRNIP1 stabilizes RAD51 on HU-induced replication arrest [29]. We therefore investigated whether WRNIP1 and RAD51 is correlated upon MRS. To this end, we performed a chromatin fractionation assay in wild-type (WSWRN) and WRN-deficient (WS) cells subjected to low-dose Aph at early or late time-points. As expected, under unperturbed conditions and after 24 hours of treatment, high salt concentration weakened the binding of WRNIP1 to chromatin in wild-type but not in WS cells (Figure 5A). Similar results were obtained at 8 hours of Aph (Figure 5A). Importantly, in WRN-deficient cells, the high levels of WRNIP1 correlated with the presence of elevated amounts of RAD51 (Figure 5A). In agreement with this observation, depletion of WRNIP1 abolished RAD51 retention in chromatin in WS cells (Figure 5B). Further supporting our hypothesis, inhibition of CHK1 activity led to an accumulation of WRNIP1 and consequently of RAD51 in wild-type cells (Figure 5C and Appendix A). By contrast, overexpression of a phospho-mimic mutant of CHK1 promoted removal of both proteins from chromatin in WS cells (Figure 5D).

Overall, our observations suggest that WRNIP1 may play a role in stabilizing RAD51 after MRS in WRN-deficient cells because of the impaired early CHK1 activation.

### 2.7. WRNIP1 Stimulates the Association of RAD51 with ssDNA in Pproximity of R-Loops upon MRS in WS Cells

WRNIP1 protects stalled replication forks from degradation, promoting RAD51 stabilization on ssDNA [29]. Hence, we first verified whether ssDNAs accumulate upon MRS in WS cells. WSWRN and WS cells were pre-labeled with the thymidine analogue 5-iodo-2’-deoxyuridine (IdU) and treated with Aph for different times as described in the scheme (Figure 6A). We specifically visualized ssDNA formation at parental-strand by immunofluorescence using an anti-BrdU/IdU antibody under nondenaturing conditions as reported [23]. Our analysis showed that WRN-deficient cells presented a significant higher amount of ssDNA than wild-type cells at later time-points of treatment (Figure 6A and Appendix A).

Next, we wondered if RAD51 localizes on parental ssDNA in a WRNIP1-dependent manner. Using a modified in situ proximity ligation assay (PLA), a fluorescence-based improved method that makes possible to detect protein/DNA association [29,38,39], we investigated the co-localization of RAD51 at/near ssDNA. To do this, WS cells depleted of WRNIP1 by RNAi were treated or not with Aph (Figure 6B). We found that the co-localization between ssDNA (anti-IdU signal) and RAD51 significantly increased after MRS in WS cells (Figure 6B). By contrast, and in agreement with the reduced levels of RAD51 in chromatin, less PLA spots were observed in the absence of WRNIP1 (Figure 6B). This result suggests that, in WS cells, the RAD51-ssDNA association requires WRNIP1 also after MRS.

It is important to note that the levels of ssDNA decreased upon ectopic expression of a phospho-mimic mutant of CHK1 in WS cells, which prevents ATM activation and retention in chromatin of WRNIP1 and RAD51 (Figure 6C and Appendix A).

As loss of WRN results in R-loop accumulation that is responsible for ATM signaling activation [18], we assessed if ssDNA could arise from persistent DNA–RNA hybrids. To prove this, we analyzed the effect of overexpression of ectopic GFP-RNaseH1, a ribonuclease that degrades RNA engaged in R-loops [40], on ssDNA formation. As can be seen in Figure 6D and Appendix A, the IdU intensity per nucleus was significantly suppressed by RNaseH1 overexpression in both the WSWRN and WS cell lines with or without treatment. Consistently, degradation of R-loops counteracted retention in chromatin of WRNIP1 and RAD51 in WS cells (Figure 6E). By contrast, preventing the processing of R-loops into double strand breaks (DSBs) by the endonuclease Xeroderma Pigmentosum complementation group G (XPG) heightened slightly the levels of parental ssDNA in WS cells (Appendix A).

To extend the above observations to the cells with defective ATR-CHK1 signaling, we evaluated direct accumulation of R-loops by immunofluorescence in WSWRN cells in which TopBP1 or Claspin was depleted. Our analysis showed that spontaneous levels of S9.6 nuclear intensity in both TopBP1- and Claspin-depleted cells were significantly higher than those observed in wild-type cells (Figure 7A). A significant enrichment of the S9.6 nuclear signal was revealed upon Aph-treatment in cells depleted for TopBP1 or Claspin with respect to wild-type counterpart (Figure 7A).

In addition, TopBP1-depleted cells accumulated ssDNA upon MRS, but treatment with 5, 6-dichloro-1-ß-D-ribofurosylbenzimidazole (DRB), an inhibitor of RNA elongation [41], led to a strong reduction of the IdU intensity per nucleus (Figure 7B and Appendix A).

Finally, to further confirm that ssDNAs are formed at/near R-loops, we performed a PLA assay. WSWRN and WS cells were incubated with Aph and DRB, then subjected to PLA using an anti-BrdU/ldU (ssDNA) and an anti-DNA-RNA hybrids S9.6 (R-loop) antibodies. Our analysis showed an increased number of spontaneous PLA spots in WS cells that were abolished by transcription inhibition (Figure 7C). After Aph treatment, PLA spots were significantly enhanced in both cell lines, with values more elevated in WRN-deficient cells, and almost completely suppressed by DRB (Figure 7C). These evidences indicate a spatial proximity between ssDNAs and R-loops.

Altogether, our findings suggest that RAD51-ssDNA association and accumulation is R-loop-dependent in WRN-deficient cells.

## 3. Discussion

Accumulation of unscheduled *R*-*loops* represents a common source of replication stress and genome instability [3,42]. Given the negative impact of aberrant R-loops on transcription, replication, and DNA repair, cells possess several mechanisms to prevent or resolve such harmful intermediates [7,42]. Currently, it is thought that an important role in avoiding deleterious consequences of these R-loops is played by some DDR proteins and replication fork protection factors [8]. Furthermore, it has recently emerged that loss of WRN, a protein involved in the repair and recovery of stalled replication forks, leads to an ATM-pathway activation to limit R-loop-associated genome instability in human cells [18]. In this study, we demonstrate that the WRN-interacting protein 1 (WRNIP1) is implicated in the response to R-loop-induced DNA damage in cells with dysfunctional replication checkpoint.

WRNIP1 is a member of the AAA+ ATPase family that was first identified as an interactor of WRN [24]. However, there is no evidence of a functional relationship between these proteins in response to MRS. Our results show that WRNIP1 coimmunoprecipitates with WRN under unperturbed conditions and that a low dose of aphidicolin slightly enhances this interaction; however, we notice that WRNIP1 is essential in the absence of WRN to counteract the effects of unscheduled R-loops. Indeed, loss of WRN or WRNIP1-depletion sensitizes human cells to Aph treatment, but concomitant lack of WRNIP1 and WRN results in a synergistic enhancement of the chromosomal aberrations frequency and DNA damage levels. These observations agree with those obtained from chicken DT40 cells that confirmed the binding of WRNIP1 to WRN, but concomitantly showed that the two proteins function independently to deal with DNA lesions during replication [25]. Of note, in yeast, deletion of Mgs1, the homolog of WRNIP1, leads to growth defects and elevated genomic instability and exhibits a relation of synthetic lethality with Sgs1, the yeast RecQ helicase [27].

WRNIP1 is stabilized in chromatin in WS cells and stabilization correlates with an inability to properly activate CHK1 upon aphidicolin, which is consistent with loss of WRN affecting ATR checkpoint activation upon MRS [18,23]. However, WRNIP1 is stabilized in chromatin also upon depletion of TopBP1, which is a key mediator of the ATR kinase [35], indicating that whenever the ATR-CHK1 signaling is dysfunctional, WRNIP1 is hyperactivated and retained stably in chromatin. In WS cells, inability to activate CHK1 early after Aph correlates to increased R-loop formation [18]. ATR-CHK1 pathway has been previously involved in safeguarding genome integrity against aberrant R-loops [43,44]. This agrees with the ability of deregulated R-loops to hamper replication fork progression [45,46,47] and also with the recent observation that depletion of ATR or CHK1 leads to R-loop-dependent replication fork stalling [43]. Consistently, and in line with other reports [21,45], we see that abrogation of essential factors for the ATR-dependent checkpoint results in high levels of R-loops. It has been previously shown that WRNIP1 is implicated in the efficient activation of ATM in response to stimuli that do not produce DNA breakage [30,48,49], and a DSB-independent but R-loop-dependent ATM pathway has been described in quiescent cells [16]. Indeed, WRN-deficient cells trigger an ATM signaling specifically after Aph-induced replication stress, which is R-loop dependent [18]. In keeping with this, we observe an R-loop-dependent hyper-phosphorylation of ATM in all the conditions tested in which the ATR checkpoint was inhibited. In addition, we find that chromatin-bound WRNIP1 is related to the late ATM-dependent CHK1 phosphorylation. Supporting this, WRNIP1 recruitment is counteracted by overexpression of a constitutively active CHK1 that, compensating for defective ATR pathway, abolishes the need to activate ATM as well as in WRN helicase dead cells that efficiently phosphorylate CHK1 [18,23]. Interestingly, degradation of R-loops weakens the association of WRNIP1 with chromatin. Hence, it is not surprisingly that, similarly, every time the replication checkpoint is compromised, WRNIP1 retention in chromatin is required for triggering an ATM-CHK1 signaling, which might be engaged in limiting transcription and/or in preventing massive R-loop-associated DNA damage accumulation. Accordingly, ATM inhibition or WRNIP1 abrogation in these pathological contexts is accompanied by increased levels of genomic instability.

Notably, combined loss of ATM activity and WRNIP1 potentiates DNA damage in cells with dysfunctional ATR checkpoint, suggesting additional roles for WRNIP1 beyond its function as a mediator of ATM. We have recently reported that WRNIP1 stabilizes RAD51 at perturbed forks after hydroxyurea treatment [29]. Of note, we observe that retention of RAD51 in chromatin in WS cells correlates with the presence of WRNIP1, and both correlate with the accumulation of R-loops. It is known that BRCA2 mediates RAD51 loading on ssDNA [50,51,52] and is required for R-loop processing [8]. WRNIP1 forms a complex with BRCA2/RAD51 [29]. It has been proposed that BRCA2 together with other proteins could contribute to preventing the collapse and reversal of R-loop-induced stalled forks, avoiding R-loop extension and promoting fork restart and R-loop dissolution [8]. In this regard, it is tempting to speculate that, in cells with replication checkpoint defects, WRNIP1 could act in concert with BRCA2 to stabilize RAD51 on ssDNA generated near/at sites of replication–transcription conflicts.

Interestingly, we observe that defective ATR checkpoint promotes accumulation of parental ssDNA that is dependent on transcription and R-loops. Such parental ssDNA is detected in proximity of R-loops upon MRS. Upon transcription–replication conflicts, exposure of parental ssDNA might derive from the unwound DNA strand at the R-loop or from the fork stalling in front of the R-loop. Hence, WRNIP1 could contribute to stabilize either RAD51 nucleofilaments assembling at the displaced DNA strand of the R-loop or at the parental DNA exposed at the fork. Of note, the fact that, upon replication–transcription collisions, WRNIP1 stimulates the stability of RAD51 at parental strand might suggest that its function is carried out before fork reversal, which is expected to result in formation of RAD51 nucleofilaments at the exposed nascent strand of the reversed arm [38,53,54]. Therefore, the WRNIP1-mediated RAD51 stabilization in chromatin might be not specific for RAD51 assembled at reversed forks, as described in hydroxyurea-treated cells [29].

Previous findings reported that the structure-specific nucleases Xeroderma Pigmentosum Complementation Group F (XPF) and XPG directly cleave R-loops to promote their resolution [55]. Mounting evidences suggest that R-loop-induced ATR activation is independent of XPG but requires the Mutagen Sensitive 81 (MUS81) endonuclease [43,49,56]. By contrast, in the absence of WRN, XPG-mediated transient DSBs deriving from R-loop processing are responsible for ATM pathway activation [18]. This agrees with previous data showing that, in WRN-deficient cells, treatment with a low dose of Aph does not induce MUS81-dependent DSB formation but determines ssDNA accumulation and enhances the number of RAD51 foci [57]. Accordingly, WRNIP1 could play two independent functions upon replication–transcription conflicts: stabilization of RAD51 at R-loops or R-loop-dependent stalled forks, and activation of ATM to repair downstream DSBs derived from the active processing of R-loops and collisions with the forks (Figure 7D).

In summary, our findings uncover a novel role of the WRNIP1-mediated response in counteracting aberrant R-loop accumulation, suggesting that a dual function of WRNIP1 is required for proper maintenance of genome stability in the pathological contexts deriving from dysfunctional ATR-dependent checkpoint. As mounting evidences reveal direct connections between R-loops and cancer [58], the elevated genome instability caused by MRS after WRNIP1 depletion in cells with dysfunctional ATR checkpoint puts forward WRNIP1 as a target to further sensitize cancer cells to inhibitors of ATR or CHK1, which are currently under clinical evaluation.

## 4. Materials and Methods

### 4.1. Cell Cultures

AG11395 (WRN-deficient) human fibroblasts retrovirally-transduced with full length cDNA encoding wild-type WRN (WSWRN) or missense-mutant form of WRN with inactive helicase (WRN^K577M^) were generated as previously described [31]. The SV40-transformed MRC5 fibroblast cell line (MRC5SV) was a generous gift from Patricia Kannouche (IGR, Villejuif, France). shWRNIP1 cell line was generated by stably expressing shRNA against WRNIP1 (shWRNIP1) (OriGene). Cells were cultured in the presence of puromycin (100 ng/mL; Invitrogen, Waltham, MA, USA) to maintain selective pressure for shRNA expression. All cell lines were maintained in DMEM (Invitrogen, Waltham, MA, USA) supplemented with 10% FBS (Boehringer Manheim-Roche, Rotkreutz, Switzerland) and incubated at 37 °C in an humified 5% CO_2_ atmosphere. All the cell lines were maintained in Dulbecco’s modified Eagle’s medium (DMEM; Life Technologies, Carlsbad, MA, USA)) supplemented with 10% FBS (Boehringer Mannheim-Roche, Rotkreutz, Switzerland) and incubated at 37 °C in a humidified 5% CO_2_ atmosphere.

### 4.2. Chromatin Fractionation

Chromatin fractionation experiments were performed as previously described with minor modifications [29]. Briefly, 1.5 × 10^7^ cells were harvested using a cell scraper, centrifuged, and then pellet was washed twice with PBS. Cell pellets were resuspended in buffer A (10 mM HEPES pH 7.9, 10 mM KCl, 1.5 mM MgCl_2_, 0.34 M sucrose, 10% glycerol, 1 mM DTT). Triton X-100 (0.1%) (Bio-rad, Hercules, California) was added, and the cells were incubated for 5 min on ice. Nuclei were collected in pellet by centrifugation. The supernatant was discarded, nuclei washed once in buffer A and then lysed in buffer B (3 mM EDTA, 0.2 mM EGTA, 1 mM DTT). Insoluble chromatin was collected by centrifugation, washed once in buffer B, and centrifuged again under the same conditions. The chromatin pellet was resuspended in buffer B diluted in cytoskeleton buffer (CSK: PIPES, 300 mM Sucrose, 3 mM MgCl_2_, and 1 mM EGTA) and then split into two equal aliquots. These samples were centrifuged, and supernatants discarded. Pellets were resuspended in buffer B/CSK supplemented with 100 mM NaCl ([low salts] extraction) and kept on ice for 10 min. Pellets containing nuclei were subjected to Western blot analysis or to further salt extraction by resuspension in buffer B/CSK supplemented with 300 mM NaCl ([high salts] extraction). After an incubation period of 10 min on ice, pellets were collected by centrifugation and supernatant was discarded. The final pellets were resuspended in 2× sample loading buffer (100 mM Tris/HCl pH 6.8, 100 mM DTT, 4% SDS, 0.2% bromophenol blue, and 20% glycerol), sonicated on ice, boiled for 5 min at 95 °C, and then subjected to Western blot as reported in Appendix A and Methods.

### 4.3. Immunofluorescence

Immunofluorescent detection of phospho-ATM was performed according to standard protocol with minor changes. Briefly, exponential growing cells were seeded onto Petri dishes, then treated (or mock-treated) as indicated, fixed in 3% formaldehyde/2% sucrose for 10 min, and permeabilized using 0.4% Triton X-100 for 10 min prior to incubation with 10% FBS for 1 h. After blocking, cells were incubated with the antibody against phospho-ATM-Ser1981 (Millipore, Burlington, Massachussets 1:300) for 2 h at RT.

To detect parental-strand ssDNA, cells were prelabeled for 20 h with 100 µM IdU (Sigma-Aldrich, Saint Louis, Missouri), washed in drug-free medium for 2 h, then treated with Aph for 24 h. Next, cells were washed with PBS, permeabilized with 0.5% Triton X-100 for 10 min at 4 °C, fixed with 3% formaldehyde/2% sucrose solution for 10 min, and then blocked in 3% BSA/PBS for 15 min as previously described [18]. Fixed cells were then incubated with anti-IdU antibody (mouse monoclonal anti-BrdU/IdU; clone b44 Becton Dickinson, Franklin Lakes, New Jersey, 1:100). The incubation with antibodies was accomplished in a humidified chamber for 1 h at RT. DNA was counterstained with 0.5 µg/ml DAPI. Images were acquired as described above.

Immunostaining for RNA–DNA hybrids was performed as described [44]. Briefly, cells were fixed in 100% methanol for 10 min at −20 °C, washed three times in PBS, pretreated with 6 μg/mL of RNase A for 45 min at 37 °C in 10 mM Tris-HCl pH 7.5 supplemented with 0.5 M NaCl, before blocking in 2% BSA/PBS overnight at 4 °C. Cells were then incubated with the anti-DNA–RNA hybrid [S9.6] antibody (Kerafast, Boston, Massachussets, 1:100) overnight at 4 °C.

After each primary antibody, cells were washed twice with PBS, and incubated with the specific secondary antibody: goat anti-mouse Alexa Fluor-488 or goat anti-rabbit Alexa Fluor-594 (Molecular Probes, Eugene, Oregon). The incubation with secondary antibodies were accomplished in a humidified chamber for 1 h at RT. DNA was counterstained with 0.5 μg/mL DAPI. Images were randomly acquired using Eclipse 80i Nikon Fluorescence Microscope (Minato, Tokyo, Japan), equipped with a VideoConfocal (ViCo) system. For each time point, at least 200 nuclei were acquired at 40× magnification. Phospho-ATM, IdU or S9.6 intensity per nucleus was calculated using ImageJ. Parallel samples either incubated with the appropriate normal serum or only with the secondary antibody confirmed that the observed fluorescence pattern was not attributable to artefacts.

### 4.4. Statistical Analysis

Statistical differences in all case were determined by two-tailed Student’s t test. In all cases, not significant; *p* > 0.05; * *p* < 0.05; ** *p* < 0.01; *** *p* < 0.001; **** *p* < 0.0001.

## 5. Conclusions

This study proposes a potential role of the WRNIP1-mediated response as regulator of R-loop-associated genomic instability. Because elevated genome instability is caused by MRS after WRNIP1 depletion in cells with dysfunctional ATR checkpoint, these findings could contribute to understanding how cells handle replication–transcription conflicts to eventually avoid early-onset of human diseases and cancer. Furthermore, as WRNIP1 has been found overexpressed in the most common cancers worldwide, such as lung and breast cancers, WRNIP1 could be considered a potential target in cancer therapy.

## Figures and Tables

**Figure 1 cancers-12-00389-f001:**
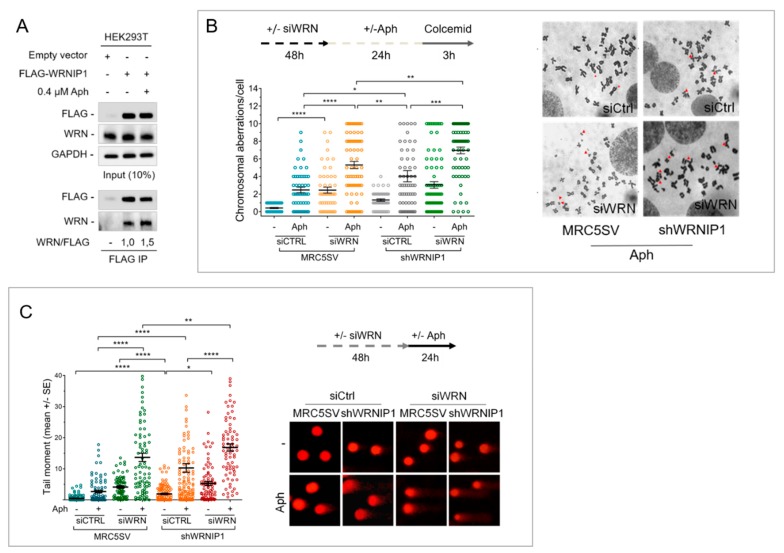
Loss of Werner Syndrome Protein (WRN) and WRNIP1 exacerbates genomic instability upon mild replication stress (MRS). (**A**) Co-IP analysis showing the interaction between WRN and WRNIP1 after IP with anti-FLAG antibody in HEK293T cell extracts, followed by Western blot with the indicated antibodies; (**B**) Analysis of chromosomal aberrations. Data are presented as chromosomal aberration per cell. Error bars represent standard error; (**C**) Alkaline comet assay in cells treated as in (**B**). Data are presented as tail moment ± SE from three independent experiments.

**Figure 2 cancers-12-00389-f002:**
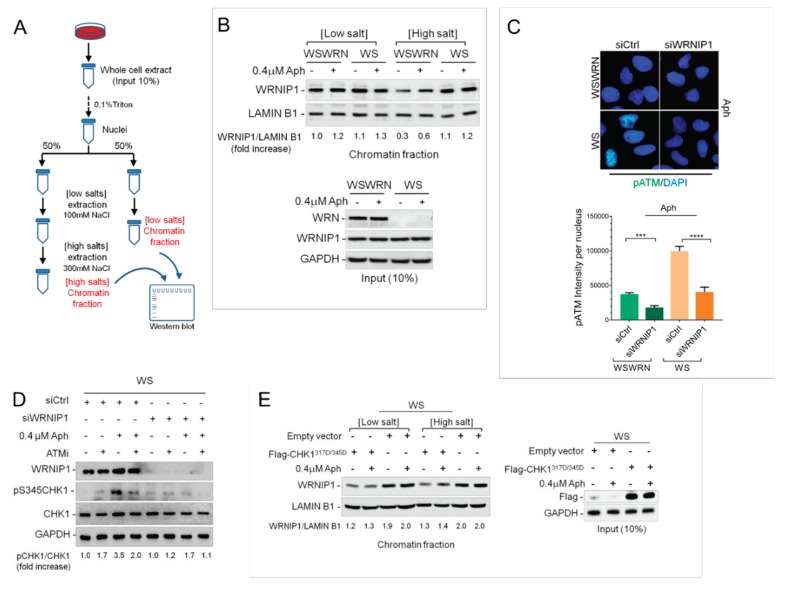
WRNIP1 mediates the ATM-dependent CHK1 phosphorylation in WS cells. (**A**) Schematic representation of chromatin fractionation assay; (**B**) The membrane was probed with the indicated antibodies. The amount the chromatin-bound WRNIP1 is reported as a ratio of WRNIP1/LAMIN B1 normalized over the untreated control; (**C**) IF analysis of cells transfected with Green Fluorecent Protein (GFP) or WRNIP1 siRNA and stained for pATM (S1981). Bar graph shows pATM intensity per nucleus. Error bars represent standard error; (**D**) WB analysis of the presence of activated, i.e., phosphorylated, CHK1 assessed using S345 phospho-specific antibody (pS345) in WS cells depleted for WRNIP1 and treated with Aph. ATMi was added 1 h prior to Aph and used as a negative control. The membrane was probed with the indicated antibodies. The normalized ratio of the phosphorylated CHK1/total CHK1 is given. (**E**) WB analysis of chromatin binding of WRNIP1, performed as in (**A**) in WS cells transfected with empty vector or FLAG-tagged CHK1317/345D and treated with Aph. The membrane was probed with the indicated antibodies. The normalized ratio of the WRNIP1/LAMIN B1 signal (chromatin) is reported.

**Figure 3 cancers-12-00389-f003:**
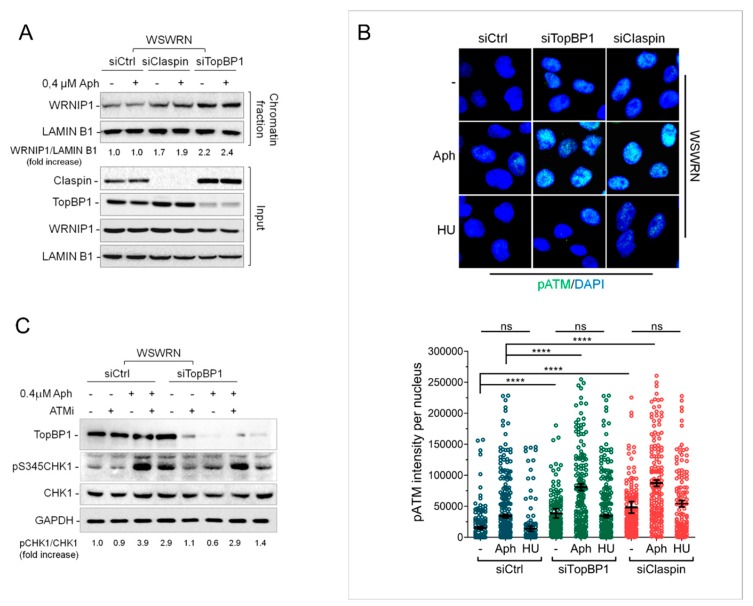
Impairment of the ATR-CHK1 signaling triggers a WRNIP1-mediated ATM activation. (**A**) WB analysis of chromatin binding of WRNIP1, performed as in Figure 2A. WSWRN cells were transfected with GFP, Claspin, or TopBP1 siRNA for 48 h and treated with Aph. The membrane was probed with the indicated antibodies. The normalized ratio of the chromatin-bound WRNIP1/LAMIN B1 is reported; (**B**) IF analysis of cells depleted of Claspin or TopBP1 as in (A) and stained for pATM (S1981). Bar graph shows pATM intensity per nucleus. Error bars represent standard error. WB using the indicated antibodies confirms depletion of Claspin and TopBP1; (**C**) WB analysis of the presence of activated, i.e., phosphorylated, CHK1 assessed using S345 phospho-specific antibody (pS345) in WSWRN cells depleted for TopBP1 as in (**A**) and treated with Aph. ATMi was added 1 h prior to Aph and used as a negative control. The membrane was probed with the indicated antibodies. The normalized ratio of the phosphorylated CHK1/total CHK1 is given.

**Figure 4 cancers-12-00389-f004:**
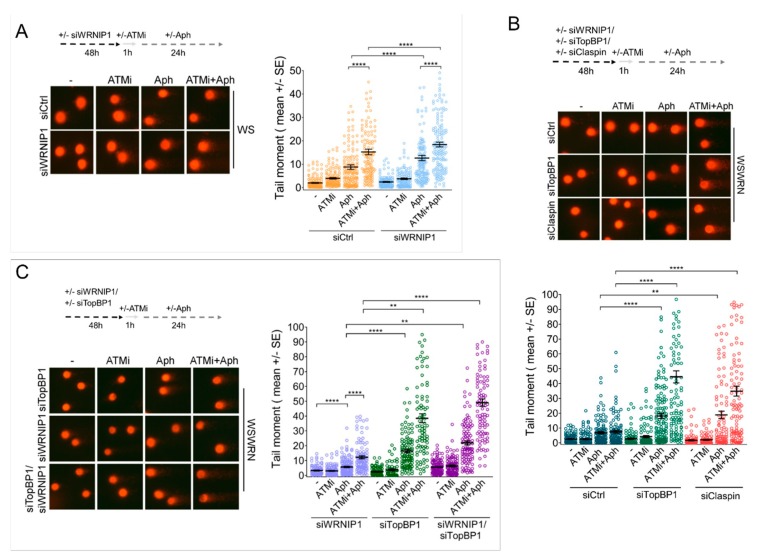
Cells with defective ATR-CHK1 signaling exhibit enhanced sensitivity to combined loss of ATM activity and WRNIP1. (**A**) Alkaline comet assay in WS cells transfected with GFP or WRNIP1 siRNA and treated as in the scheme. Data are presented as tail moment ± SE from three independent experiments; (**B**) Alkaline comet assay in WSWRN cells after TopBP1 or Claspin depletion in WSWRN cells as reported in the scheme. Data are presented as tail moment ± SE from three independent experiments; (**C**) Alkaline comet assay in WSWRN cells after TopBP1 and/or WRNIP1 depletion as reported in the scheme. Data are presented as tail moment ± SE from three independent experiments.

**Figure 5 cancers-12-00389-f005:**
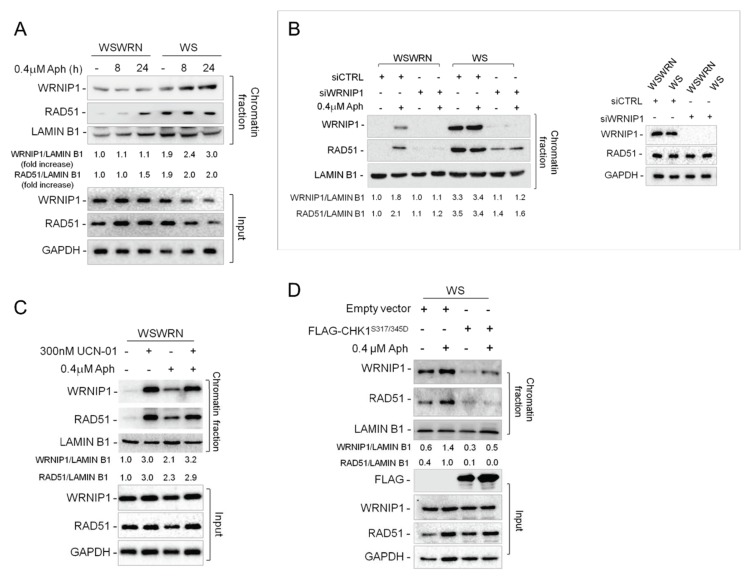
WRNIP1 stabilizes RAD51 on chromatin in WS cells. (**A**) WB analysis of chromatin binding of WRNIP1 and RAD51, performed as in Figure 2A, in WSWRN and WS cells treated with Aph for different times. The membrane was probed with the indicated antibodies. The normalized ratio of the chromatin-bound WRNIP1/LAMIN B1 or RAD51/LAMIN B1 are given; (**B**) WB analysis of chromatin binding of WRNIP1 and RAD51, performed as in Figure 2A, in WSWRN and WS cells transfected with GFP or WRNIP1 siRNA treated with Aph for 24 h. The membrane was probed with the indicated antibodies. The normalized ratio of the chromatin-bound WRNIP1/LAMIN B1 or RAD51/LAMIN B1 are given; (**C**) WB analysis of chromatin binding of WRNIP1 and RAD51, performed as in Figure 2A, after CHK1 inhibition. WSWRN cells were treated with Aph, and/or to 300 nM of CHK1 inhibitor, UCN-01, for the last 6 h. The membrane was probed with the indicated antibodies. The normalized ratio of the chromatin-bound WRNIP1/LAMIN B1 or RAD51/LAMIN B1 are given; (**D**) WB analysis of chromatin binding of WRNIP1 and RAD51, performed as in Figure 2A. WS cells were transfected with empty vector or FLAG-tagged CHK1^317/345D^ and treated with Aph. The membrane was probed with the indicated antibodies. The normalized ratio of the chromatin-bound WRNIP1/LAMIN B1 or RAD51/LAMIN B1 are given.

**Figure 6 cancers-12-00389-f006:**
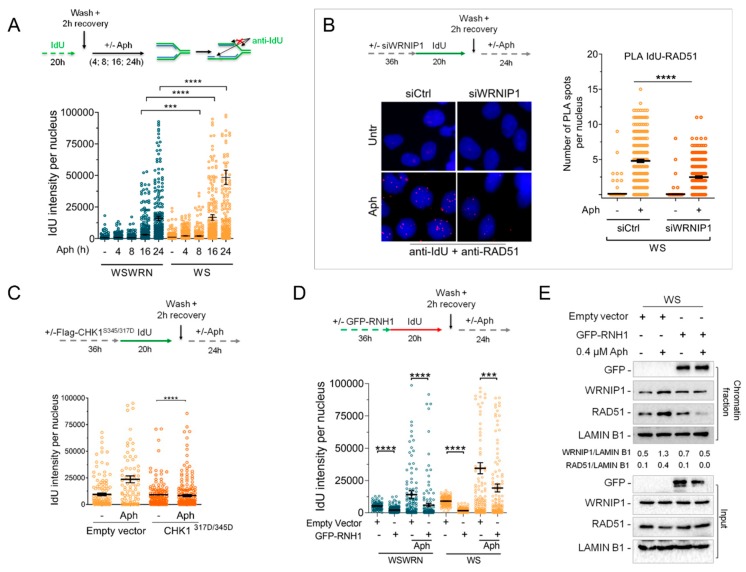
RAD51-ssDNA association requires WRNIP1 in WS cells upon MRS. (**A**) Detection of ssDNA by IF with an anti-BrdU/IdU antibody in WSWRN and WS cells treated with 0.4 µM Aph for different times as reported in the scheme. Graph shows the mean ± SE from three independent experiments; (**B**) Analysis of proximity between ssDNA (anti-BrdU/IdU antibody) and endogenous RAD51 (anti-RAD51 antibody) performed by in situ PLA assay in WS cells depleted of WRNIP1 and treated as reported in the scheme. PLA interaction is shown in red. Graph shows the number of PLA spots per nucleus; (**C**) Detection of ssDNA by IF with anti-BrdU/IdU antibody in WS cells transfected with an empty vector or a FLAG-tagged CHK1^317/345D^ and treated with 0.4 µM Aph. Graph shows the mean ± SE from three independent experiments; (**D**) Detection of ssDNA by IF with anti-BrdU/IdU antibody in WSWRN or WS cells treated as reported in the scheme after transfection with a plasmid expressing GFP-tagged RNaseH1. Graph shows the mean ± SE from three independent experiments; (**E**) WB analysis of chromatin binding of WRNIP1 and RAD51 in WS cells treated as in (**D**). Chromatin fractionation was performed as described in Figure 2A. The membrane was probed with the indicated antibodies. The normalized ratio of the chromatin-bound WRNIP1/LAMIN B1 or RAD51/LAMIN B1 are given.

**Figure 7 cancers-12-00389-f007:**
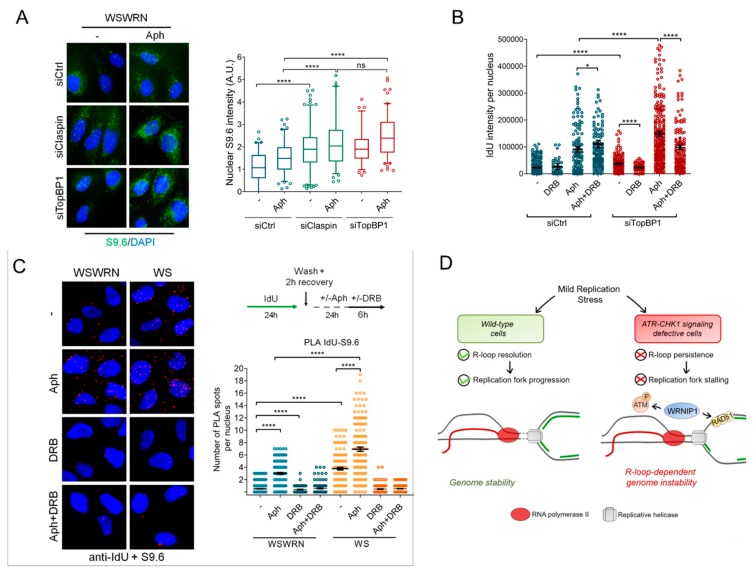
Spatial proximity between ssDNA and R-loops upon MRS in cells with dysfunctional ATR-dependent checkpoint. (**A**) Immunofluorescence detection of R-loops with S9.6 antibody in WSWRN cells transfected with GFP, Claspin, or TopBP1 siRNA and treated with 0.4 µM Aph for 24 h. Box plot shows nuclear S9.6 fluorescence intensity. Box and whiskers represent 20–75 and 10–90 percentiles, respectively; (**B**) detection of ssDNA by IF with an anti-BrdU/IdU antibody in WSWRN transfected with GFP or TopBP1 siRNA and treated with 0.4 µM Aph for 24 h. Graph shows the mean ± SE from three independent experiments; (**C**) analysis of proximity between ssDNA (anti-BrdU/IdU antibody) and R-loops (anti-S9.6 antibody) performed by in situ PLA assay in WSWRN cells transfected with GFP or TopBP1 siRNA and treated as reported in the scheme. PLA interaction is shown in red. Graph shows the number of PLA spots per nucleus; and (**D**) proposed model of a role of WRNIP1 in suppressing R-loop-induced genome instability in cells with dysfunctional ATR-mediated checkpoint.

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
