# Peer review of "Checkpoint Defects Elicit a WRNIP1-Mediated Response to Counteract R-Loop-Associated Genomic Instability"

_cancers, 2020, doi:10.3390/cancers12020389_

Round 1
Reviewer 1 Report
This manuscript examines the impact of Werner helicase interacting protein 1 (WNRIP1)’s loss on replication stress and genome instability. The headline novel assertion in the paper is that WNRIP1 suppresses R-loop-driven genome instability, which is evident when ATR-mediated checkpoint response is dysfunctional.
The several experiments are well-executed and the conclusions relevant for cancer biology and R-loop fields.
I think that the paper is suitable for publication in its present form after minor changes as follow:
-WRNIP1 identity should be introduced since the beginning in the abstract.
-line 12: I would not say that “how cells handle these structures (R-loops) is not fully understood” rather that more factors involved in R-loop resolution perhaps are not known.
-line 37: “Apart of canonical mechanisms…” it is not clear what is the difference between canonical and non-canonical mechanisms that resolve R-loops. Please clarify this.
-line 299: “…arguing against an end-resection-related origin”. This sentence is also not clear. As stated by authors in the discussion, other nucleases can process R-loops into DSBs.
-line 356: Hrq1 is another yeast RecQ helicase (see for example Rogers et al, 2017, NAR, 45:5217-5230).
-Some infos about treatments must be missing above in Figure 1 panel A.
-Standardize the font size of the text (see for example lines 47, 122 and 355).
Author Response
Answer to Reviewer #1s comments:
This manuscript examines the impact of Werner helicase interacting protein 1 (WNRIP1)’s loss on replication stress and genome instability. The headline novel assertion in the paper is that WNRIP1 suppresses R-loop-driven genome instability, which is evident when ATR-mediated checkpoint response is dysfunctional. The several experiments are well-executed and the conclusions relevant for cancer biology and R-loop fields.
We thank very much the reviewer for the interest in our finding, for her/his appreciation of our study and for the constructive comments.
I think that the paper is suitable for publication in its present form after minor changes as follow:
-WRNIP1 identity should be introduced since the beginning in the abstract.
According to the reviewer’s suggestion, we have introduced the WRNIP1 identity in the abstract.
-line 12: I would not say that “how cells handle these structures (R-loops) is not fully understood” rather that more factors involved in R-loop resolution perhaps are not known.
We agree with the reviewer’s observation, and we have amended accordingly in the revision version of the manuscript trying to better clarify the concept.
-line 37: “Apart of canonical mechanisms…” it is not clear what is the difference between canonical and non-canonical mechanisms that resolve R-loops. Please clarify this.
We have amended the sentence accordingly in the revised version of the manuscript to better clarify the concept.
-line 299: “…arguing against an end-resection-related origin”. This sentence is also not clear. As stated by authors in the discussion, other nucleases can process R-loops into DSBs.
We agree with the reviewer's observation and amended the text accordingly.
-line 356: Hrq1 is another yeast RecQ helicase (see for example Rogers et al, 2017, NAR, 45:5217-5230).
We have amended accordingly the revised version of the manuscript.
-Some infos about treatments must be missing above in Figure 1 panel A.
We are sorry for this inconvenient, we have replaced the Figure 1 panel A with the right one.
-Standardize the font size of the text (see for example lines 47, 122 and 355).
We have checked the font size of the text in the revised version of the manuscript.
Reviewer 2 Report
In this manuscript the authors evaluate the role of WRNIP1 in limiting genome instability under mild replication stress conditions in compromised ATR mediated checkpoint cells such as WRN deficient cells. Here, the authors demonstrate that WRNIP1 is stabilised in chromatin in WRN deficient cells and provide data that implicates WRNIP1 in the response to R-loop-induced DNA damage in cells with dysfunctional replication checkpoint. This manuscript is largely an extension of the authors previous work published in EMBO journal titled “WRNIP1 protects stalled forks from degradation and promotes fork restart after replication stress”. It is well written, appropriately cited and methods well described.
Minor concerns:
To demonstrate that WRNIP1 mediated ATM activation is DSB indepdent, can the authors show DSB markers such as H2AX and 53BP1 by immunofluorescence and/or western blotting ?
Does the genome instability upon loss of WRN and WRNIP1 lead to cell death after MRS ? What happens in cell viability assays?
Author Response
Answer to Reviewer #2s comments:
In this manuscript the authors evaluate the role of WRNIP1 in limiting genome instability under mild replication stress conditions in compromised ATR mediated checkpoint cells such as WRN deficient cells. Here, the authors demonstrate that WRNIP1 is stabilised in chromatin in WRN deficient cells and provide data that implicates WRNIP1 in the response to R-loop-induced DNA damage in cells with dysfunctional replication checkpoint. This manuscript is largely an extension of the authors previous work published in EMBO journal titled “WRNIP1 protects stalled forks from degradation and promotes fork restart after replication stress”. It is well written, appropriately cited and methods well described.
We thank the reviewer for her/his appreciation of our study and for the constructive comments.
Minor concerns:
To demonstrate that WRNIP1 mediated ATM activation is DSB indepdent, can the authors show DSB markers such as H2AX and 53BP1 by immunofluorescence and/or western blotting ?
Two earlier studies have reported a DSB-independent ATM activation: one in wild-type cells in response to low-dose of Aphidicolin (Kanu et al., Oncogene, 2016) and the other in quiescent cells (Tresini et al., Nature, 2015). The latter work showed a DSB-independent but R-loop-dependent ATM activation. Interestingly, we have recently demonstrated that transient DSBs resulting from R-loop processing by XPG endonuclease are responsible for ATM signalling activation in response to MRS in WRN-deficient cells (Marabitti et al., NAR, 2019). In the present study, which is in part an extension of our NAR paper, we prove that WRNIP1 is implicated in triggering the ATM pathway upon MRS in WRN-deficient cells and, more in general, in cells with checkpoint defects. Therefore, these findings allow us to assume that transient DSBs are responsible for the WRNIP1-mediated ATM activation upon MRS in cells with checkpoint defects.
Does the genome instability upon loss of WRN and WRNIP1 lead to cell death after MRS ? What happens in cell viability assays?
For what concerns the exacerbated genomic instability detected in shWRNIP1 cells in which WRN is downregulated by RNA interference, we think that it is plausible that it could result in cell death. Although the short time provided to send revision of our manuscript did not allow us to perform the actual experiment, using a fluorescence-based cell viability assay (LIVE/DEAD assay), we previously established that inhibition of ATM in WRN-deficient cells increases cell death upon MRS (Marabitti et al., NAR, 2019). Similarly, upon depletion of WRNIP1, which responsible for the ATM signalling activation, WRN-deficient cells might undergo extensive cell death after MRS.